# AMPK-Mediated Regulation of Alpha-Arrestins and Protein Trafficking

**DOI:** 10.3390/ijms20030515

**Published:** 2019-01-25

**Authors:** Allyson F. O’Donnell, Martin C. Schmidt

**Affiliations:** 1Department of Biological Sciences, University of Pittsburgh, Pittsburgh, PA 15260, USA; 2Department of Microbiology and Molecular Genetics, University of Pittsburgh, Pittsburgh, PA 15219, USA

**Keywords:** *Saccharomyces cerevisiae*, Snf1 kinase, AMP-activated protein kinase, α-arrestins, arrestin-domain containing proteins, glucose transporters, 2-deoxyglucose, ubiquitination, NEDD4

## Abstract

The adenosine monophosphate-activated protein kinase (AMPK) plays a central role in the regulation of cellular metabolism. Recent studies reveal a novel role for AMPK in the regulation of glucose and other carbohydrates flux by controlling the endocytosis of transporters. The first step in glucose metabolism is glucose uptake, a process mediated by members of the GLUT/SLC2A (glucose transporters) or HXT (hexose transporters) family of twelve-transmembrane domain glucose transporters in mammals and yeast, respectively. These proteins are conserved from yeast to humans, and multiple transporters—each with distinct kinetic properties—compete for plasma membrane occupancy in order to enhance or limit the rate of glucose uptake. During growth in the presence of alternative carbon sources, glucose transporters are removed and replaced with the appropriate transporter to help support growth in response to this environment. New insights into the regulated protein trafficking of these transporters reveal the requirement for specific α-arrestins, a little-studied class of protein trafficking adaptor. A defining feature of the α-arrestins is that each contains PY-motifs, which can bind to the ubiquitin ligases from the NEDD4/Rsp5 (Neural precursor cell Expressed, Developmentally Down-regulated 4 and Reverses Spt- Phenotype 5, respectively) family. Specific association of α-arrestins with glucose and carbohydrate transporters is thought to bring the ubiquitin ligase in close proximity to its membrane substrate, and thereby allows the membrane cargo to become ubiquitinated. This ubiquitination in turn serves as a mark to stimulate endocytosis. Recent results show that AMPK phosphorylation of the α-arrestins impacts their abundance and/or ability to stimulate carbohydrate transporter endocytosis. Indeed, AMPK or glucose limitation also controls α-arrestin gene expression, adding an additional layer of complexity to this regulation. Here, we review the recent studies that have expanded the role of AMPK in cellular metabolism to include regulation of α-arrestin-mediated trafficking of transporters and show that this mechanism of regulation is conserved over the ~150 million years of evolution that separate yeast from man.

## 1. Introduction

Cells must sense and respond to the carbohydrate supply in their environment to ensure continued energy supply. The preferred carbohydrate fuel for most eukaryotic cells is glucose, and its uptake from the environment is controlled by a family of facilitative glucose transporters known as GLUTs in mammalian cells or HXTs (hexose transporters) in *Saccharomyces cerevisiae* [1,2,3]. Once inside the cell, glucose—through the concerted actions of glycolysis, the citric acid cycle, and oxidative phosphorylation—is used to generate a reservoir of adenonsine triphopshpate (ATP), the cellular energy currency. ATP in turn drives the vast majority of energy-requiring activities in the cell. This cascade of events highlights the critical importance of glucose uptake; without it, generation of ATP becomes inefficient in some cell types and a shortage in ATP supply can cause the energy-dependent reactions in the cell to grind to a halt. Indeed, dysregulation of glucose uptake is associated with a wide array of metabolic disorders, the most prevalent of which is diabetes with >30 million Americans diagnosed with this disorder (see CDC report 2017 https://www.cdc.gov/diabetes/pdfs/data/statistics/national-diabetes-statistics-report.pdf).

Given the interconnectedness of glucose uptake and the ATP energy balance in the cell, it makes intuitive sense that the cell coordinates these two processes. At the heart of this regulatory circuit lies the adenosine monophosphate-activated protein kinase (AMPK in mammals, Snf1 in *Saccharomyces cerevisiae*), which is conserved across eukaryotes and becomes an active kinase when the AMP/ATP or ADP/ATP ratios in the cell increase [4]. Elevated adenosine monophosphate (AMP) and adenosine diphosphate (ADP) acts as a molecular red flag, signaling that the cell may not be able to meet future cellular energy demands. In response, AMPK stimulates catabolic pathways needed to bolster ATP reserves while slowing cellular anabolic processes to reduce energy demands [5]. A key outcome of AMPK activation in response to falling energy levels is increased glucose uptake [6,7,8,9,10]. To achieve this end, AMPK alters the trafficking of glucose transporters in multiple ways. For example, in muscle cells AMPK phosphorylates TBC1 domain family member 1D (TBC1D), a RAB-GTPase activating protein (GAP) that inhibits fusion of glucose transporter isoform 4 (GLUT4) storage vesicles with the plasma membrane, to impair its function and so stimulate fusion of GLUT4-containing vesicles with the plasma membrane and improve glucose uptake [11,12]. A similar stimulation of glucose transporter isoform 1 (GLUT1) translocation to the plasma membrane occurs upon AMPK activation in hamster kidney cell lines [13]. In contrast, in rat liver cells AMPK can stimulate GLUT1-mediated glucose uptake in a translocation-independent fashion [9,10], a phenomenon first described almost 20 years ago; however, the mechanism underlying this AMPK-dependent activity remains unknown.

Excitingly, more recent studies have defined a new role for AMPK in regulating the protein trafficking of glucose and other carbohydrate transporters. These studies provide a long sought-after mechanistic answer to how AMPK stimulates glucose uptake in the absence of glucose transporter translocation [14,15]. In brief, AMPK has been shown to phosphorylate and inhibit multiple members of a protein trafficking adaptor family, known as the α-arrestins, which are responsible for the endocytosis of multiple carbohydrate transporters. The α-arrestins, conserved across eukaryotic lineages with the exception of plants, act as cargo selective trafficking adaptors that bind to both transmembrane proteins and a ubiquitin ligase [16]. By bringing the ubiquitin ligase into close proximity with select membrane proteins (referred to hereafter as “cargos”), α-arrestins facilitate ubiquitination of the membrane cargo, which serves as a mark for cargo incorporation into endocytic vesicles [17,18,19,20,21]. Herein we review a complementary suite of studies from the budding yeast *Saccharomyces cerevisiae* and mammalian cell lines. These studies have converged on a remarkably similar regulatory model whereby phosphorylation of the α-arrestins by AMPK prevents α-arrestin-mediated endocytosis of glucose and other carbohydrate transporters (Figure 1) [15,17,22]. In addition to regulating α-arrestin function, AMPK further exerts control over the expression of α-arrestins in mammalian cells, and possibly analogous carbon-source-dependent expression changes are also reported for yeast α-arrestins (Figure 2) [23,24,25,26]. Together, these studies reveal an ancient mechanism for sensing and responding to cellular energy status that predates the evolution of multicellularity and has important clinical ramifications for the treatment of human metabolic disorders. 

## 2. The Arrestin Family of Protein Trafficking Adaptors

In 2008, the α-arrestins were first identified as part of the larger arrestin family, which spans the Spo0M-related proteins in bacteria, Vps26-related proteins found in all branches of life, the broadly conserved α-arrestins and the most recently evolved β- and visual-arrestins (the latter two referred to hereafter collectively as β-arrestins) [27]. The crystal structures for several family members have been solved identifying conserved N- and C-terminal arrestin-fold domains, each of which contains 7 anti-parallel beta-sheets connected by linker regions of variable length and composition [28,29,30,31]. These arrestin-folds are the defining feature of the family. β-arrestins, the most well-studied class of arrestins, operate as multi-faceted protein trafficking adaptors that bind to membrane cargo proteins, including G-protein coupled receptors, and interact with the AP-2 adaptin complex and clathrin to promote endocytic turnover of their cognate cargos [32,33]. β-arrestins control post-endocytic sorting of their membrane cargos and can also act as signaling scaffolds, binding to multiple protein kinases and thereby tethering their activity to a specific cellular location [33]. Phosphorylation of the β-arrestins is often associated with inhibition of their endocytic function [34,35,36]; however, to our knowledge AMPK has not been show to phosphorylate β-arrestins directly.

Unlike their β-arrestin relatives, functions for α-arrestins have only more recently been described and studies of the α-arrestins are closer to their inception. The α-arrestins are conserved from yeast to man; to date there are 14 α-arrestins identified in yeast (Rod1/Art4, Rog3/Art7, Aly1/Art6, Aly2/Art3, Ldb19/Art1, Csr2/Art8, Ecm21/Art2, Rim8/Art9, Art5, Art10, Bul1, Bul2, Bul3, and Spo23) and 6 in mammalian cells (Arrestin-Domain Containing Protein (ARRDC)1, ARRDC2, ARRDC3, ARRDC4, ARRDC5, and Thioredoxin Interacting Protein 1 (TXNIP)). Many of the initial studies defining α-arrestin function in protein trafficking come from work in *S. cerevisiae* [17,18,19,20,37]. These studies demonstrate that α-arrestins function in an analogous fashion to the β-arrestins; α-arrestins bind selectively to membrane cargo proteins and help stimulate their endocytosis in a signal-dependent manner [17,18,19,20,37]. A distinguishing feature of the α-arrestins is the presence of ^L^/_P_PxY motifs in the C-terminal tail that extends from the C-terminal arrestin-fold domain. In yeast, these motifs are required for α-arrestin interaction with the WW-domains of the ubiquitin (Ub) ligase Rsp5, a member of the mammalian NEDD4 Ub ligase family. Almost all α-arrestins in yeast have been shown to bind to Rsp5 and mutation or deletion of the ^L^/_P_PxY motifs typically renders the α-arrestin non-functional [18], unable to stimulate the trafficking of their known cargo proteins. Analogously, association of mammalian α-arrestins with Nedd4-family ubiquitin ligases has been shown to be required for ARRDC protein trafficking activity [38,39]. The membrane cargos targeted for α-arrestin regulation are a large and diverse group (Table 1 and Table 2), including G-protein coupled receptors, amino acid permeases, metal ion transporters and carbohydrate transporters [17,18,19,22,40,41,42].

α-Arrestin-mediated protein trafficking is regulated by posttranslational modifications including ubiquitination and phosphorylation [17,18,20,22,37,40,43,44,49,54,55,64,65]. Phospho-regulation of α-arrestins is particularly complex, with multiple phosphorylated residues identified in high-throughput or directed mass spectroscopy (MS) analyses [37,43,66,67,68]. For α-arrestins Aly1 and Ldb19, 24 and 13 phosphorylated residues were identified in targeted MS approaches, respectively [37,43]. The high degree of phospho-modification on α-arrestins is suggestive of a potentially complex combinatorial control acting to fine tune α-arrestin function. Many of the phosphorylated residues identified to date cluster in the C-terminal tail of the α-arrestin, which is predicted to be structurally flexible, or within predicted unstructured pockets in the N-terminal arrestin-fold domain ([37,43]; O’Donnell and Schmidt unpublished observation). In yeast, multiple kinases, including Snf1 (the yeast ortholog of mammalian AMPK), Npr1, Pho85, Yck1/Yck2 (the yeast ortholog of mammalian casein kinase 1) and Ypk1 (the yeast ortholog of mammalian Sgk1/2 kinases), are capable of directly phosphorylating the α-arrestins [20,37,40,55,69,70], and many more kinases have been shown to have catalytic activity towards the α-arrestins in high-throughput in vitro kinase assays [71,72]. The activity of the protein phosphatases calcineurin (type 3 protein phosphatase), Glc7 (type 1 protein phosphatase) and Sit4 (type2-related protein phosphatase) have been shown to either directly or indirectly regulate the phosphorylation and/or activity of α-arrestins [17,40,43,46,54,73,74].

The functional ramifications of α-arrestin phosphorylation are dependent upon the specific phosphorylation event examined [17,20,37,40,43,44,49,54,55,75]. However, an emerging theme in the phospho-regulation of α-arrestins is that phosphorylation typically impairs α-arrestins’ endocytic trafficking function while dephosphorylation of the α-arrestin facilitates or improves α-arrestin-mediated endocytosis [17,22,37,43]. The phosphorylated form of the α-arrestin may be important for intracellular sorting functions that are starting to be described for the α-arrestins in shuffling proteins between intracellular compartments, such as the endosome and Golgi [20,64,76]. This inhibition of α-arrestin function by phosphorylation is conserved in the studies of mammalian Txnip, where AMPK or AKT phosphorylation of Txnip impedes its endocytic activities [14,15]. The mechanisms underlying the AMPK-mediated inhibition of both yeast and mammalian α-arrestins are discussed in detail below.

## 3. AMPK-Regulation of α-Arrestin-Mediated Trafficking in *S. cerevisiae*

In yeast, the AMPK heterotrimer is comprised of: Snf1, the catalytic α subunit; Snf4, the γ subunit; and one of three β subunit isoforms—Gal83, Sip1 or Sip2 [77]. These regulatory β subunit isoforms localize to distinct cellular compartments, suggesting that they may play a role in sequestering the function of the Snf1 kinase, and their localization is dynamically altered in response to stress conditions that activate the kinase [78,79]. In yeast, as in mammalian cells, Snf1 acts as a sensor of the energy and nutritional status in the cell. Activated Snf1 is phosphorylated on threonine 210 in response to a variety of environmental changes, including low glucose, alkaline or heat shock stressors [80,81,82]. Perhaps the best-characterized roles of activated Snf1 is to phosphorylate the transcriptional activators Cat8 and Sip4 [83,84,85] and transcriptional repressors Mig1 and Mig2 [86,87,88,89,90]. This in turn leads to the gene expression alterations required for cells to adapt to the select stress agents [91]. This regulatory role is thought to be analogous to the phosphorylation of PCG-1α and FoxO1 by AMPK in mammalian cells [92,93]. Recently, multiple α-arrestins—including Rod1, Rog3, and Csr2—have been shown to be direct substrates of the Snf1 kinase [94], and still other α-arrestins have been identified as in vitro substrates of Snf1 in high-throughput analyses—including Aly2 and Bul2 [71]. Snf1 phosphorylation of α-arrestins has been shown to inhibit α-arrestin-mediated trafficking; however, the mechanism of this inhibition varies for each α-arrestins and may be dependent upon the mode of activation for Snf1. Here we describe the findings that demonstrate a direct role for Snf1 in controlling the α-arrestin-mediated protein trafficking of multiple carbohydrate transporters and also in the regulation of G-protein coupled receptor (GPCR) function in yeast.

### 3.1. AMPK Inhibits α-Arrestin-Mediated Trafficking of Glucose Transporters

In 2007, Rod1 was first described as an arrestin and demonstrated to be a direct substrate of the Snf1 kinase complex in vitro [94], and this finding has been further supported by subsequent in vitro analyses [40] (Schmidt, unpublished observations). In vivo, mutation of serine 447 in Rod1 to alanine abolished the Snf1-dependent mobility shift of Rod1, suggesting that this residue—just downstream of the C-terminal arrestin-fold domain—is a site of Snf1 regulation [94]; however, the functional ramifications of this modification were not yet clear. Further in vitro assays identified 6 serine residues—Ser315, Ser447, Ser641, Ser706, Ser720 and Ser781—as required for phosphorylation of Rod1 by Snf1 [40]. Mutation of these 6 serines to alanines, mimicking the dephosphorylated form of Rod1, improved Rod1-mediated adaptation to the α-factor pheromone, which is a readout of activation of the G-protein coupled receptor Ste2 controlled mating pathway in yeast [40]. The fact that loss of Snf1-dependent phospho-sites improves Rod1 function suggests that Rod1 is negatively regulated by Snf1 in this pathway and could have implications for Rod1-mediated control of Ste2 trafficking.

The first evidence that Rod1 could act as an endocytic adaptor was a study that demonstrated Rod1 stimulation of the endocytosis of the high affinity glucose transporter Hxt6 [19], although this study did not explore the regulation of Rod1 by the Snf1 kinase. However, a subsequent study suggests that Snf1 kinase activity may impede Rod1 association with Hxt6 in vivo [65], but the mechanism of this regulation remains to be defined. Snf1-dependent, α-arrestin-mediated regulation of the low-affinity glucose transporters Hxt1 and Hxt3 has also been described. Hxt1 and Hxt3 are predominantly localized to the cell surface in yeast grown in standard 2% glucose-containing medium. Under these energy-replete conditions, the Snf1 kinase is not robustly activated. However, addition of the toxic glucose analog, 2-deoxyglucose (2DG), can activate the Snf1 kinase by promoting its phosphorylation at threonine 210 [95]. Even under glucose replete conditions, the presence of small quantities of 2DG is toxic to cells, and cells lacking the Snf1 kinase are even more sensitive to 2DG than their wild-type counterparts [95]. Surprisingly, over-expression of Hxt1 or Hxt3 suppresses the sensitivity of *snf1*∆ cells, spurring localization studies of these glucose transporters. In the absence of Snf1, Hxt1 and Hxt3 abundance at the plasma membrane is reduced, and this effect is further exacerbated by the addition of 2DG [22]. Excitingly, the α-arrrestin Rod1, and to a lesser extent its paralog Rog3, are required for the 2DG-induced endocytosis of Hxt1 and Hxt3. Consistent with a role for Rod1 and Rog3 downstream of Snf1, loss of Rod1 and Rog3 restores 2DG-resistance to *snf1*∆ cells back to wild-type levels. To regulate the 2DG-stimulated endocytosis if Hxt1 and Hxt3, Rod1 and Rog3 must interact with Rsp5, suggestive of a role for the α-arrestins in controlling ubiquitination of these glucose transporters (Figure 1a).

Consistent with the earlier in vitro findings, Rod1 and Rog3 are phosphorylated in a Snf1-dependnet manner and this phosphorylation is further promoted by addition of 2DG [22]. Interestingly, the Snf1-dependent phosphorylation of Rod1 precluded ubiquitination of the α-arrestin. Mono-ubiquitination of α-arrestins is associated with activation of their endocytic function. Phospho-inhibition of α-arrestin ubiquitination can therefore be a proxy for impaired trafficking activity. This is similar to the Snf1-induced phospho-inhibition of Rod1 ubiquitination described below for Rod1-mediated trafficking of Jen1 [65]. However, in this instance, we do not yet know if Rod1 association with the 14-3-3 proteins is altered as a result. Thus, our current model is that Snf1-dependent phosphorylation of Rod1 prevents its ubiquitination and impairs Rod1-mediated endocytosis of Hxt1 and Hxt3. 

The mechanism of action for Snf1 phospho-inhibition of Rog3 appears to be dramatically different. In the presence of 2DG, Rog3 is also phosphorylated in a Snf1-depedent manner, which is likely inhibitory to Rog3-mediated protein trafficking of Hxts. However, when Snf1 is hyperactivated, as is the case in cells lacking the protein phosphatase Glc7/Reg1 complex needed to dephosphorylate and deactivate Snf1, Rog3 protein levels are very low [22]. When Snf1 is additionally deleted from cells lacking Reg1, Rog3 is stabilized. These findings demonstrate that Rog3 is phosphorylated in a Snf1-dependent manner and that this phosphorylation can result in loss of Rog3 stability. Deciphering the molecular mechanism underlying this Rog3 destabilization will undoubtedly be of great future interest to the cell biology community, as this regulator network has striking similarities to that demonstrated for AMPK-mediated control of Txnip [15], which has important ramifications in the treatment of metabolic diseases.

### 3.2. AMPK Inhibits α-Arrestin-Mediated Trafficking of the Jen1 Lactate Permease and Other Membrane Cargo

The first study that linked Snf1 to the trafficking activity of an α-arrestin focused on the role of Rod1 in glucose-induced degradation of the lactate transporter Jen1 [17], which is a member of the monocarboxylate SLC16/MCT family of transporters [96]. In marked contrast to the low-affinity Hxt1 and Hxt3 glucose transporters described above, in cells grown on glucose, the Jen1 lactate permease is internalized and degraded; in cells grown on lactate, Jen1 is stabilized at the cell surface, where it imports lactate as a carbon source. Growth of cells on lactate, which is not a preferred carbon source, likely activates the Snf1 kinase. Glucose-induced internalization of the Jen1 permease is partially impaired, and the post-endocytic sorting of Jen1 to the vacuole (yeast equivalent of the lysosome) is also impeded in the absence of Rod1 [17,46,76]. Rod1 acts redundantly in the glucose-induced endocytosis of Jen1 with the α-arrestin Bul1, and together the two adaptors are required for glucose-induced endocytosis of Jen1 [46]; however, Bul1 has not been shown to be regulated by Snf1. It has been shown that the glucose-induced trafficking of Jen1 requires the ubiquitin ligase Rsp5; mutation of the ^L^/_P_PxY motifs in Rod1 needed for its interaction with Rsp5 impairs Jen1 trafficking to the vacuole in response to glucose [17]. Deletion of Rod1 also reduces the glucose-stimulated ubiquitination of Jen1, which is a precursor to Jen1 sorting to the vacuole [17]. Interestingly, in cells grown on lactate as a carbon source, conditions where the Jen1 permease is stabilized at the cell surface, Rod1 is highly phosphorylated, and this phosphorylation depends on the Snf1 kinase [17]. As stated above, Snf1 itself is activated by phosphorylation at threonine 210, and dephosphorylation of this site by type 1 protein phosphatase (PP1) in yeast (comprised of the Glc7 catalytic subunit and the Reg1 regulatory subunit) inactivates Snf1. As a result, in cells lacking Reg1, Snf1 is hyper-activated and Rod1 becomes hyper-phosphorylated, irrespective of the carbon source employed [80]. Hyper-phosphorylated Rod1 is unable to regulate trafficking of Jen1 to the vacuole.

From these studies, it is clear that Snf1-mediated phosphorylation of Rod1 impairs Rod1-mediated trafficking of the Jen1 permease to the vacuole. Furthermore, Rod1 phosphorylation in response to growth on lactate medium results in its association with the 14-3-3 proteins, Bmh1 and Bmh2 [17]. Phosphorylation and 14-3-3 protein interaction impede Rod1 ubiquitination, which is a modification needed to activate α-arrestin-mediated trafficking [17]. Thus, from these studies we can propose a model whereby Snf1 phosphorylation of Rod1 results in Rod1 binding to 14-3-3 proteins, impairing Rod1 ubiquitination and resulting in a trafficking-incompetent Rod1 that fails to simulate the degradative trafficking of Jen1 (Figure 1b) [17,46,76].

Most recently, phosphorylation of α-arrestin Bul1 has been shown to be regulated in response to glucose starvation [46]. Loss of the Sit4 phosphatase augments Bul1 phosphorylation; however, it is unclear if Sit4 is acting directly to dephosphorylate Bul1 under these conditions or if it may dephosphorylate and therefore inactivate Snf1 [97,98]. Bul1 is also important for the glucose-induced trafficking of Jen1 to the vacuole. It is tempting to speculate that Bul1 regulation maybe analogous to that described above for Rod1 in these conditions. It will be interesting to see in future studies if Bul1 is also targeted by the Snf1 kinase, expanding the functional connection between α-arrestins and AMPK.

## 4. AMPK- and Carbon-Source Regulation of α-Arrestin Gene Expression

In mammalian cells, the arrestin family is expanded to include both α-arrestins and β-arrestins. The mammalian α-arrestins are comprised of ARRDC1-5 and TXNIP/VDUP1 [27], while the β-arrestins include both the visual arrestins SAG and ARR3 and the β-arrestins ARRB1 and ARRB2. β-arrestins have well-described and important roles in controlling cellular metabolism (reviewed in Zhao and Pei [99]), including regulation of insulin receptor signaling [100] and insulin-resistance [101]. However, to date β-arrestins have not been shown to be phosphorylated by AMPK and so we will not describe their roles further here. Three α-arrestins—TXNIP, ARRDC3 and ARRDC4—have been reported to regulate glucose and lipid homeostasis in mammalian cells and/or mouse models [102,103,104,105]. Interestingly, some of this regulation is gender specific, with increased expression of ARRDC3 in males from Icelandic populations linked to obesity and increased expression of ARRDC3 in omental adipose correlated with obesity in males ([102] and references therein). In a mouse model, loss of ARRDC3 expression protects against obesity and increases energy use by improving the thermogenesis of both brown and white fat tissues [102]. Thus, the mechanism underlying this ARRDC3-driven metabolic alteration is not thought to be linked to glucose uptake, but rather is due to a change in the β3-adrenergic receptor activity, nor is it known to be associated with AMPK-regulation [102,103].

In marked contrast, the activities of TXNIP and ARRDC4 are tightly connected to glucose uptake, and the expression and activity of these two α-arrestins are regulated in response to glucose supply. TXNIP is the most highly induced gene in pancreatic β-islet cells in response to glucose and its expression is upregulated in mouse models of diabetes [26,106,107,108]. More recently, ARRDC4 expression was also shown to be induced by glucose treatment of β-islet cells [109]. Both TXNIP and ARRDC4 expression are under the glucose-stimulated control of the ChREBP/MondoA family of transcription factors. Phospho-regulation of these transcription factors is controlled by AMPK [25]. Here we will describe the current state of our understanding of AMPK-mediated control of α-arrestin transcription, which relies on ChREBP/MondoA, and also discuss the carbohydrate-responsive transcriptional changes observed for α-arrestins in yeast.

### 4.1. Control of Txnip Expression by MondoA and ChREBP

The identification of TXNIP as a key target of glucose-induced transcription predates the structural studies that defined the α-arrestin family and that conclusively define N- and C-terminal arrestin-fold domains in TXNIP [26,27,31]. Therefore, early studies of TXNIP do not refer to it as a member of the α-arrestin protein family. TXNIP is named for its ability to interact with and inhibit thioredoxin, which is a critical regulator of cellular redox potential. However, binding and inhibition of thioredoxin are not common features in α-arrestins, with TXNIP being the only α-arrestin known to bind this ligand to date [110,111]. While there are thioredoxin-associated functions for TXNIP (reviewed in [112]), TXNIP interaction with thioredoxin may not be required for its role in regulating glucose metabolism as mutations of TXNIP at cysteine 247, which is required for TXNIP association with thioredoxin, have no impact on TXNIP-mediated inhibition of glucose uptake in adipocytes [31,103]. Rather, two major players in the glucose-induced expression of the α-arrestin TXNIP are the MondoA/ChREBP transcription factors and AMPK.

The carbohydrate response element binding protein (ChREBP and also known as MondoB) and MondoA are a related pair of basic helix-loop-helix-leucine zipper (bHLHZ) transcription factors that bind to carbohydrate response elements in the promoters of target genes [113,114]. These transcription factors heterodimerize with Mlx, a shared interacting partner required for their role as transcriptional activators. Together ChREBP and MondoA are responsible for most the glucose-induced transcriptional changes across an array of tissue types, including β-islet cells, liver, heart and skeletal muscle [23,115] and reviewed in [113]. A key target of ChREBP and MondoA activity is the TXNIP gene (Figure 2). The TXNIP promoter contains a tandem copy of the E-box element (sequence 5’CACGTG3’), which is the consensus for a carbohydrate response element bound by ChREBP and Mondo A, and TXNIP is the highest transcriptionally activated gene upon glucose addition to β-islet cells [23,26]. TXNIP gene expression is upregulated by ChREBP in rodent β-islet cells and hepatocytes [23,116], while MondoA drives TXNIP expression in human β-islet cells, kidney cells and skeletal muscle cells [105,109,117].

The activity of ChREBP and MondoA is regulated by their localization; ChREBP and MondoA reside in puncta in the cytosol or on the mitochondrial surface, respectively, in limiting glucose conditions but relocalize to the nucleus in response to glucose addition [105,118,119]. The glucose-sensing factor needed to stimulate this localization change is somewhat controversial, with some reports suggesting that glucose-6-phosphate is important for this translocation while others demonstrate that 3-O-methylglucose, which is not metabolized by cells, is able to stimulate ChREBP/MondoA-induced gene expression [117,120,121,122]. In each of these studies, TXNIP expression was specifically monitored as a readout of ChREBP or MondoA activity, and in many of them, activation of TXNIP expression is linked to reduced glucose uptake.

Phosphorylation is a potent and complex regulator of ChREBP/MondoA nuclear translocation and/or its DNA binding activity. It has been proposed that under low-glucose conditions, protein kinase A (PKA) phosphorylates ChREBP, and that in response to glucose, protein phosphatase 2A (PP2A) dephosphorylates it at serine 196 and threonine 666; dephosphorylation stimulates its nuclear localization and, in turn, its transcriptional activation function [119]. However more recent reports demonstrate that mutation of these phosphorylation sites does not alleviate the glucose-responsive function of ChREBP [124,125] and suggests that additional layers of phospho-regulation are critical for ChREBP function. Further studies have revealed multiple phosphorylation sites from distinct signaling pathways converge on ChREBP to generate complex combinatorial control of this transcription factor [113]. In contrast to glucose activation of ChREBP, increased fatty acid levels—as occurs in rats fed a high-fat diet—activates AMPK in the liver and stimulates AMPK-mediated phosphorylation of ChREBP at serine 568 [25]. The broader function of AMPK in response to a high-fat diet has previously been reviewed [126]. AMPK phosphorylated ChREBP fails to activate gene expression, which could therefore dampen TXNIP expression. Consistent with this model for AMPK-mediated inhibition of TXNIP expression, treatment of β-islet cells with free fatty acids, which acts in concert with low glucose to signal a starvation state, increases AMPK activity and decreases TXNIP expression in a ChREBP-dependent manner. In the presence of free-fatty acids, ChREBP is excluded from the nucleus and TXNIP expression is repressed [123]. Indeed, even in the presence of glucose, modest residual AMPK function impedes TXNIP expression, as knockdown of the AMPK-α1 catalytic subunit augmented TXNIP expression [123]. In sum, these experiments suggest a model whereby AMPK, either robustly or modestly activated by elevated free fatty acids or the glucose-fed state, phosphorylates ChREBP preventing induction of TXNIP gene expression (Figure 2b). This serves as an added layer of AMPK-mediated inhibition, on top of the AMPK-dependent inhibition of TXNIP-mediated protein trafficking (see Section 6, below).

The control of TXNIP expression by AMPK is more complex than this model would suggest. TXNIP can induce its own expression in a feedforward loop and this process is inhibited by AMPK [127]. TXNIP overexpression in rat β-islet cells promotes ChREBP dephosphorylation at serine 196 and subsequent nuclear translocation. Since TXNIP is a direct target for ChREBP, this stimulation of ChREBP by TXNIP represents a form of autoactivation of gene expression. Activation of AMPK, by addition of AICAR, inhibited TXNIP expression, demonstrating that AMPK plays an antagonistic role. Surprisingly, TXNIP overexpression impedes AMPK activation, as reduced phospho-AMPK was observed in cells over-expressing TXNIP [127]. The downstream factor that controls TXNIP-mediated inhibition of AMPK is not yet defined, but could be due to inhibition of the AMPK-activating kinases or activation of phosphatase that dephosphorylates AMPK. Thus, the interplay between TXNIP and AMPK is a complex one, with these two proteins each imposing regulation on the other at multiple levels. 

### 4.2. Control of ArrDC4 Expression by MondoA

High glucose levels not only activate transcription of TXNIP, but also stimulates expression of α-arrestin ARRDC4 in human β-islet cells and this activation is dependent upon MondoA [109]. Phosphorylation plays a role in regulating this activity; however, to date it is PKA regulation that appears to be important rather than AMPK. Activated PKA by treatment with forskolin prevents MondoA nuclear localization, blocking ARRDC4 expression. Consistent with a role for MondoA in controlling ARRDC4 expression, in human embryonic kidney epithelial cells (HA1ER) glucose refeeding after starvation stimulates ARRDC4 transcription [117]. These findings suggest that glucose and ChREBP/MondoA regulated expression of α-arrestins may be global regulatory features. Indeed, in cultured bovine granulosa cells, a model for luteinization during folliculogenesis in ovulation, cells are grown to a high density and their transcriptional profiles mimic those observed during folliculogenesis. In this model, TXNIP and ARRDC4 gene expression are among the most highly down-regulated genes, suggesting that an analogous ChREBP and/or MondoA pathway may exist in this cell type, consistent with the increased glucose uptake observed for hormone-stimulated follicle cells [128].

### 4.3. Altered Expression of Yeast α-Arrestins in Response to Carbon Supply

Carbon source switching, and specifically shifts in glucose abundance, also alters α-arrestin transcription in *S. cerevsiae*. Specifically, expression of α-arrestin Csr2 in yeast is induced by a shift to lactate as a carbon source, which is thought to mimic glucose starvation conditions. In yeast cells grown in glucose replete conditions, Csr2 gene expression is inhibited with almost no Csr2 transcript detected ([24] and Schmidt, unpublished observations). However, when cells are shifted into lactate-containing medium, a condition known to activate the Snf1 kinase [85], Csr2 gene expression increases dramatically as the transcriptional repressors Mig1 and Mig2 are phospho-inhibited by Snf1-mediated phosphorylation and now fail to bind the Csr2 promoter [24]. Under these conditions, the endocytic function of Csr2 is active and the protein is ubiquitinated at lysine 670. This ubiquitination is needed for Csr2-mediated endocytosis of the high-affinity glucose transporter Hxt6. When cells are shifted into glucose-replete medium, Csr2 transcription is repressed and the protein becomes phosphorylated by PKA, which either impedes its ubiquitination or stimulates its deubiquitination, thereby blocking its endocytic function [24]. In contrast to the regulation observed for Csr2, expression and abundance of the α-arrestin Rod1 is very high in glucose grown cells and is repressed when cells are shifted into lactate-containing medium (Figure 2a). Expression and activity of Rod1 under glucose replete conditions is analogous to the expression changes observed for TXNIP. However to date the factors needed for this expression change are not defined in yeast and, while there are bHLH transcription factors of the Myc/Max family in yeast, and many of them are important regulators of nutrient balance [129], evolutionary analysis has yet to reveal a yeast homolog for ChREBP and MondoA [130]. It has been postulated that yeast may contain as-yet-undefined members of the bHLHZ family, as they contain E-box sequences and other interacting partners for this family [131]. Given the strong parallels in other regulatory features, it will be interesting to see if transcriptional activators are revealed in future studies of α-arrestin gene expression that could act analogously. 

## 5. AMPK Regulation of α-Arrestin-Mediated Trafficking in Mammals

The fact that AMPK activation increases glucose uptake in skeletal muscle, kidney, adipose and liver cells is well documented [6,7,9,10,13,132,133]. In muscle cells, AMPK activity increases the amount of the Glut4 transporter at the cell surface by stimulating translocation of Glut4-containing storage vesicles (GSV) to the plasma membrane [11,12]. However, AMPK also stimulates GSV-trafficking independent increases in glucose uptake. Specifically, AMPK stimulates GLUT1-mediated glucose uptake in a translocation-independent fashion [9,10]. The mechanism underlying this AMPK- and GLUT1-controlled glucose uptake became clear when AMPK was shown to inhibit TXNIP-regulated endocytosis of GLUT1 in liver cells [15]. This study, aimed initially at understanding the ChREBP transcriptional control of TXNIP (described above in Section 4), identified a slower migrating form of TXNIP on immunoblots from cells starved for glucose or treated with the glucose analog, 2DG. This slower migrating band is the result of TXNIP phosphorylation by AMPK, as the band is more prominent in extracts from cells treated with AMPK activators or 2DG and was lost in cells lacking AMPK or when extracts were treated with phosphatase. In vitro assays demonstrate that AMPK phosphorylates TXNIP directly on serine 308 and this phosphorylation not only inhibits TXNIP-mediated endocytosis of GLUT1, which is compellingly demonstrated by co-localization of TXNIP with clathrin-coated vesicles using total internal reflection fluorescence microscopy and measurements of GLUT1 internalization from the cell surface, but also increases the rate of TXNIP degradation [15]. These findings provide the molecular underpinnings for the long-standing question in the field of how AMPK activity can stimulate glucose uptake; the inhibition of the TXNIP-mediated endocytosis of the glucose transporter GLUT1 closes this regulatory circuit (Figure 1c).

This phospho-inhibition of TXNIP-regulated protein trafficking also occurs in liver and breast cancer cell lines that are treated with growth factor; however, the kinase responsible in this case is not AMPK, but is rather protein kinase B (AKT) [14]. AKT phosphorylates TXNIP at serine 308, the same site as AMPK, and this phosphorylation reduces the phospholipid binding of TXNIP, in turn impeding its ability to be recruited to cellular membranes and impact protein trafficking [14]. Consistent with the model of AMPK-regulation of TXNIP, phosphorylation of TXNIP by AKT in adipocytes similarly blocks TXNIP-mediated endocytosis of GLUT4.

Overall, the AMPK and AKT phospho-inhibition of TXNIP in mammalian cells shares considerable similarities with the AMPK phospho-regulation of α-arrestins Rod1 and Rog3 in yeast (Figure 1). AMPK targets an analogously located phospho-site in the c-terminal tail of Rod1 as that described for TXNIP. However, additional phosphorylation sites in Rod1 and Rog3 also seem to be controlled by AMPK. Phosphorylation impairs the endocytic trafficking of both TNXIP and Rod1. However, to our knowledge, AMPK-phosphorylated Rod1 is not destabilized upon phosphorylation [22]. The yeast paralog of Rod1, Rog3 is also phosphorylated in an AMPK-dependent manner and this phosphorylation destabilized the Rog3 protein (O’Donnell and Schmidt, unpublished). It is tempting, therefore, to speculate that this phosphorylation-controlled degradation of Rog3 is analogous to that observed for TXNIP. Additionally, though it is impossible to determine if Rod1/Rog3 and TXNIP are orthologous, these α-arrestins from yeast seem functionally analogous to TXNIP in mammalian cells. It will be intriguing to see if Rod1 and/or Rog3 have any additional TXNIP-associated functions, such as binding thioredoxin or controlling mitochondrial dynamics, in the yeast model. It will also be interesting to see if the phosphorylation of TXNIP results in 14-3-3 binding, as this is a rather common mode of α-arrestin inhibition in yeast [17,24,44,49]. Rod1 and Rog3 require the ubiquitin ligase Rsp5 to mediate glucose transporter trafficking. However, the ubiquitin ligase needed for TXNIP-mediated trafficking of mammalian GLUTs is yet to be defined. Given that other mammalian α-arrestins operate in concert with members of the NEDD4 family of ubiquitin ligases, the orthologs of yeast Rsp5, it is likely that this family is involved in TXNIP function. Indeed, AMPK-mediated regulation of the epithelial sodium transporter requires NEDD4-2 in an oocyte system providing an additional mechanism by which AMPK can regulate protein trafficking [134].

## 6. Conclusions and Future Directions

The studies summarized here reveal an exciting and highly conserved regulatory circuit linking AMPK to α-arrestin-mediated trafficking of glucose and other carbohydrate transporters (Figure 1). In broad strokes, activation of AMPK/Snf1 results in phospho-inhibition of α-arrestin-mediated trafficking of the GLUT/Hxt family of glucose transporters in both yeast and mammalian cells. AMPK phosphorylation impairs the ability of α-arrestins to stimulate endocytosis of glucose transporters, resulting in increased glucose uptake. In mammalian cells, AMPK-inhibition of α-arrestins also occurs at the level of transcription, where AMPK phosphorylation of the ChREBP/MondoA transcription factors impairs their localization to the nucleus, precluding transcriptional activation of TXNIP or ARRDC4 (Figure 2). The ChREBP/MondoA transcription activators and the expression of yeast and mammalian α-arrestins are highly responsive to the glucose status in cells. Interestingly, the Snf1 kinase plays a key role in altering the expression of α-arrestins, activating the expression of Csr2 in response to growth on lactate as a carbon source. This is the opposite of the effect observed for AMPK-mediated repression of TXNIP and ARRDC4. However, what maintains expression of α-arrestin Rod1 under replete glucose conditions in yeast? Rod1 expression could be analogously controlled by a pathway akin to the ChREBP/MondoA regulatory circuit that regulates TXNIP. 

It will be interesting in the future to see if application of these current paradigms for AMPK-mediated control of α-arrestins can provide new mechanistic understanding to long-standing questions of AMPK-regulated uptake of glucose and other nutrients. For example, AMPK is a potent regulator of glucose uptake in the kidney; however, it is unclear what links AMPK to the glucose transporters in this system (reviewed in [135]). Are α-arrestins acting downstream of AMPK activation in this system to regulate glucose transporter availability at the cell surface? In the small intestine, GLUT2 activity at the cell surface is increased when AMPK is activated as well [136,137]; however, the details underlying this increased GLUT2 activity are not well described. Could α-arrestins be negatively regulated by AMPK in the intestine to help stimulate glucose uptake via GLUT2? It will be exciting to see the answer to these questions in the future and we anticipate the emergence of a more global role for AMPK-mediated regulation of α-arrestins and protein trafficking.

## Figures and Tables

**Figure 1 ijms-20-00515-f001:**
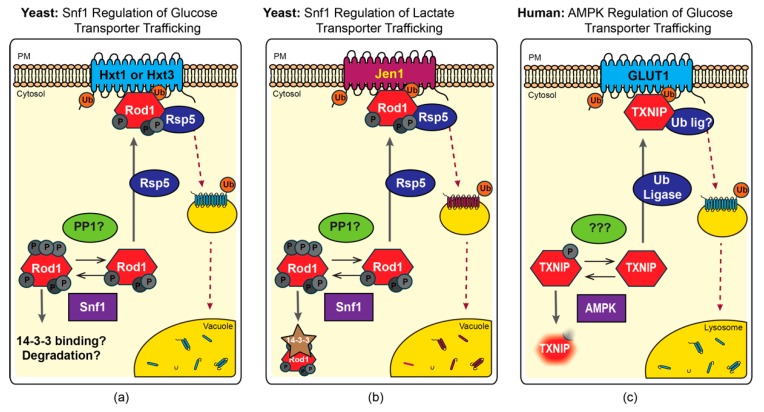
Snf1/AMPK-mediated regulation of membrane transporter trafficking in yeast and humans. (**a**) Hyper-phosphorylation of yeast α-arrestin Rod1 by Snf1 kinase inhibits the ability of Rod1 to promote ubiquitination, endocytosis and degradation of hexose transporters 1 and 3 (Hxt1 and Hxt3) [22]. Hyper-phosphorylation of Rod1 may result in 14-3-3 binding and/or Rod1 degradation. (**b**) Hyper-phosphorylation of yeast α-arrestin Rod1 by Snf1 kinase sequesters Rod1 in a complex with 14-3-3 proteins and inhibits the ability of Rod1 to promote ubiquitination, endocytosis and degradation of lactate transporter Jen1 [76]. (**c**) Phosphorylation of human α-arrestin TXNIP promotes its degradation, thereby inhibiting its ability to promote endocytosis and degradation of glucose transporter GLUT1 [15]. The ubiquitin ligase involved in this process has yet to be defined. Grey arrows indicate pathway connections and red dashed arrows indicated protein trafficking events.

**Figure 2 ijms-20-00515-f002:**
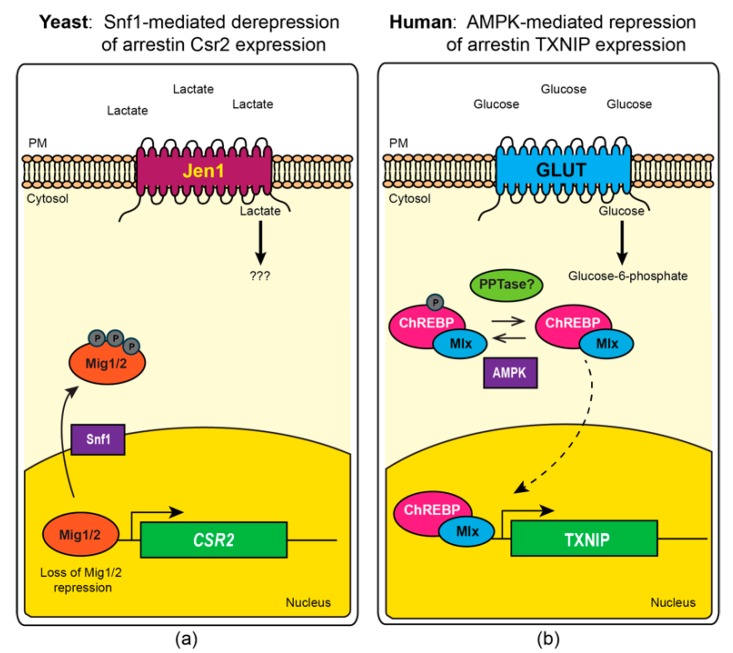
Snf1/AMPK-mediated regulation of gene expression in yeast and human. (**a**) Snf1 kinase-mediated phosphorylation of the yeast transcriptional repressors Mig1 and Mig2 promotes their translocation out of the nucleus (denoted by solid black arrow) leading to derepression of α-arrestin Csr2 [24]. (**b**) AMPK-mediated phosphorylation of human transcriptional activator ChREBP blocks its ability to induce expression of α-arrestin TXNIP [123]. Upon dephosphorylation, ChREBP-Mlx can translocate into the nucleus to activate expression of TXNIP (denoted by dashed black line).

**Table 1 ijms-20-00515-t001:** Yeast α-arrestins and their known cargos.

α-Arrestin Genes and Cargos in Yeast			
Gene	Alias	Cargo	References
*ALY1*	*ART6*	Gap1, Dip5, Ste3	[20,41,43,44]
*ALY2*	*ART3*	Gap1, Dip5, Ste3	[20,41,43,44,45]
*ART5*		Itr1	[19]
*ART10*		No known cargo	
*BUL1*	*SMM2, DAG1, RDS1*	Jen1, Gap1, Ptr2, Tat1, Tat2, Ctr1, Put4, Dal5	[46,47,48,49,50,51,52]
*BUL2*		Gap1, Ptr2, Tat1, Tat2, Ctr1, Put4, Dal5	[48,49,50,51,52]
*BUL3*		No known cargo	
*CSR2*	*ART8, MRG19*	Hxt6, Hxt7, Hxt2, Hxt4	[24,53]
*ECM21*	*ART2*	Tat2, Fur4, Lyp1, Smf1	[19]
*LDB19*	*ART1*	Mup1, Ste2, Ste3, Can1, Lyp1, Tat2, Fur4	[19,41,54]
*RIM8*	*ART9*	Rim21, Pma1	[42,55,56]
*ROD1*	*ART4*	Hxt1, Hxt3, Hxt6, Jen1	[17,19,22]
*ROG3*	*ART7*	Hxt3	[22]
*SPO23*		No known cargo	

**Table 2 ijms-20-00515-t002:** Mammalian α-arrestins and their known cargos.

α-Arrestin Genes and Cargos in Humans			
Gene	Alias	Cargo	References
*ARRDC1*		YAP1, Notch, TSG101, DMT1	[57,58,59,60]
*ARRDC2*		No known cargo	
*ARRDC3*		YAP1, PAR1, β3-AR, β2-AR, V2R, ITG β4	[39,57,61,62,63]
*ARRDC4*		MDA5, DMT1, V2R, β2-AR	[60,62]
*ARRDC5*		No known cargo	
*TXNIP*	*VDUP1*	GLUT1, GLUT4	[14,15]

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
