# Peer review of "AMPK-Mediated Regulation of Alpha-Arrestins and Protein Trafficking"

_ijms, 2019, doi:10.3390/ijms20030515_

Reviewer 1 Report

Alpha arrestins are adapter proteins that shuttle the ubiquitin ligases (e.g. Rsp5 in yeast or Nedd4 in mammals) to transporters or receptors at the plasma membrane but also to intracellular proteins to mediate their ubiquitination that causes degradation or changes in activity. As various carbohydrate transporters are targeted by a-arrestins, this mechanism was suggested to contribute to the changes in transporter expression in situations when changes in the carbohydrate hydrate source occur. As AMP-activated protein kinase (AMPK) is activated by intracellular energy depletion which initially occurs if the carbohydrate source will not be uptaken, it is suggested that this kinase contributes to this a-arrestin function. This review summarizes the current knowledge of the impact of AMPK on activity and expression of a-arrestins in yeast and mammals.

Major comments:

The overall outline of the manuscript should be changed. At various sites the authors state: “as discussed above” or “as discussed below”. One way to improve the structure and thereby facilitate the reading is first to discuss the function of the a-arrestins in yeast and mammals for the transporters and then discuss the expression regulation in both organisms. Linking the paragraphs which explain the mechanisms with the respective figures will help the authors to reorganize the structure of the text as the figures should be placed in close proximity of their discussion.

Some of the discussed topics, like the thioredoxin binding of TXNIP (line 303 to 310) in a section of expression control of Txnip is misleading at the place mentioned and in this detail out of the focus of this review.

As the regulation of a-arrestins is not the only mechanism to regulate ubiquitination in an AMPK-dependent manner, a short paragraph on the ubiquitin ligase Nedd4 as a direct target of AMPK should be included.

The fact that b-arrestins are not a target of AMPK appear twice in the text. Delete the second mentioning (around line 277).

The term “phosphor-regulation” should be avoided as it is lab slang.

In the legend of Figure 1a needs to be clearer and more straightforward in a similar way as the legend for 1b and 1c.

Is there any hint why in the presence of glucose phosphorylation of Rod1 primarily inhibits association with Rsp5 whereas in the presence of lactate Rod1 phosphorylation causes association with 14-3-3 proteins and subsequent degradation?

The description of the mechanism from line 203 to line 235 needs thorough re-editing: Small mistakes in the grammar make the understanding difficult (e.g. line 204 “it” instead of “its”. Wording like “Surprisingly” “Excitingly” etc should be avoided. Also the difference of Rog3 to Rod1 regulation needs to be written much clearer. In the present form the reader needs to read this paragraph several times to understand the “dramatic” difference.

Minor comments:

Line 30: should read 150 million years. At the end of the line delete one “.”

Line 78: delete the term “plot-thickening twist”.

Line 181: Define what sort of cargo Ste2 is.

Line 350: It is unclear what the authors mean with “complex combinatorial control”?

Line 360 to 364: In one sentence the term “activating” appears 4 times. Please change as this is not a good style.

At most sites in the text the phosphorylated amino acids are fully written but at some places (e.g. line 254) the abbreviation is used. Please be consistent within the text.

The references need to be uniform. Most titles of the publications contain no capitals, but some (e.g. Ref 96) do. Again please be consistent.

Author Response

Response to Reviewer 1:

1. Reviewer 1 felt that the “overall outline of the manuscript should be changed”.

Since the other two reviewers explicitly state that the manuscript was “informative and instructive” and “well-written”, we do not think a large-scale change to the organization at this stage is advisable. It would not necessarily improve the manuscript for all readers and would unnecessarily delay publication.

2. Reviewer 1 felt that our discussion of TXNIP binding to thioredoxin was “out of the focus of this

review.”

This section is relevant because TXNIP was discovered and named for its ability to bind thioredoxin. That is worthy of discussion when we introduce this protein and explain that it has a second role as an arrestin. This further highlights a problem in the field whereby the literature describing TXNIP function in binding thioredoxin ignores and rarely refers to the protein as an arrestin and the findings that describe the protein trafficking function for TXNIP sometimes fail to discuss its role in binding thioredoxin. The two are undoubtedly linked and we therefore would like to keep this paragraph as written.

3. Reviewer 1 requested the addition of the AMPK-mediated regulation of NEDD4.

We added a few sentences (lines 473-479) and reference a paper from Ken Hallows group showing that NEDD4-2 is needed for AMPK-mediated regulation of ENaC. (Ho 2018 JBC; ref 133).

4. Reviewer 1 notes that we mention twice that b-arrestins are not a target of AMPK and suggests we

delete the second mentioning (around line 277).

Since the b-arrestins are the more well-known family members with better described functions, it is important to distinguish them from the a-arrestins, which are the focus of this review. We would like to keep both mentions of this interesting distinction between the a- and b-arrestins.

5. Reviewer 1 did not like the use of the term “phosphor-regulation”.

I was not able to find this in the draft we submitted or in the edited draft we were asked to revise. The legend to Fig 1a was simplified and clarified as requested.

6. Why does Rod1 respond differently in glucose and lactose?

In the presence of glucose, Rod1 exists in a balance of phosphorylated and dephosphorylated forms. The dephosphorylated forms would presumably no longer be bound to 14-3-3 proteins, and therefore no longer inhibited, however this has not been demonstrated experimentally. This gives rise to the modest internalization of glucose transporters in glucose-grown cells. It may also explain the endocytosis of Jen1 observed in glucose-grown cells, as some of the arrestin is ‘active’ under this condition. In the presence of 2DG, Rod1 becomes ubiquitinated and appears to be more active as an endocytic adaptor, since we have increased removal of glucose transporters from the cell surface. Presumably there would be more Rsp5 bound and increased dissociated from 14-3-3 proteins for Rod1 under these conditions, however this has not been tested experimentally. In the presence of lactose, Rod1 is largely phosphorylated, dissociates from Rsp5 and is bound by the inhibitory 14-3-3 proteins. In this scenario, Rod1 is no longer competent to stimulate Jen1 trafficking to the vacuole. In short, growth in glucose vs lactate give rise to two very different energy statuses in the cell. AMPK responds to this and alters the phosphorylation pattern on Rod1. In lactate, Rod1 is more completely phosphorylated by AMPK (and AMPK activity is likely higher). In glucose, Rod1 is a mixed population of phosphorylated and dephosphorylated forms, which could give rise to a small population of active arrestin needed for controlling both Jen1 and Hxt transporter endocytosis. It should be noted that cells acutely shifted from lactate to glucose display a different pattern of post-translational modification, based on electrophoretic mobility shift changes. However, to date no quantitative comparative MS analyses has been done to map the sites of post-translational modifications or the altered binding affinity of Rod1 for Rsp5 and/or 14-3-3 proteins under these two conditions. The residues targeted by AMPK under these two conditions may be different.

7. is vs it’s.

This has been corrected: “by promoting its phosphorylation” (current draft line 209).

8. Reviewer 1 did not like our use of words such as surprisingly and excitingly.

This is a matter of personal style and does not affect scientific content. No changes.

9. Line 30: should read 150 million years. At the end of the line delete one “.”

Changed billion to million. Deleted one ‘.’

10. Line 78: delete the term “plot-thickening twist”.

Changed to “In addition to regulating a-arrestin function,”

11. Line 181: Define what sort of cargo Ste2 is.

Ste2 is a G-protein coupled receptor as is stated in the sentence.

12. Line 350: It is unclear what the authors mean with “complex combinatorial control”?

This means that multiple signaling pathways converge on this transcription factor. This sentence was changed to “Further studies have revealed multiple phosphorylation sites from distinct signaling pathways converge on ChREBP to generate complex combinatorial control of this transcription factor.”

13. Line 360 to 364: In one sentence the term “activating” appears 4 times. Please change as this is not a good style.

The offending sentence was changed to “In sum, these experiments suggest a model whereby AMPK, either robustly or modestly activated by elevated free fatty acids or the glucose-fed state, phosphorylates ChREBP preventing induction of TXNIP gene expression” thus reducing the use of the word “activating” from four to one.

14. At most sites in the text the phosphorylated amino acids are fully written but at some places (e.g. line 254) the abbreviation is used. Please be consistent within the text.

Changed two occurrences of Thr210 to threonine 210.

15. The references need to be uniform. Most titles of the publications contain no capitals, but some (e.g.Ref 96) do. Again please be consistent.

This is really an issue for the copy editor. Nonetheless, we manually edited reference 96.

Reviewer 2 Report

This review describes the regulation of alpha-arrestins and protein trafficking by AMPK in S cerevisiae and in mammals. This comprehensive review is very nicely written and well-focused on regulation and function of alpha-arrestins and AMPK. It is informative and instructive both for researchers in this field and for broader people in the biomedical fields. The narrative has several innovating and even thought-provoking ideas and brings us up to date with the latest advances. There are only a few general comments.

1. There was numerous studies and a nice review by Ruderman et.al (JCI, 2013) suggesting a reduction in AMPK activity in obesity/diabetes models such as in leptin deficient ob/ob mice or high fat fed mice. Interestingly, mice fed on a high carbohydrate diet exhibited increased AMPK activity/expression in the liver (Loh et.al Hepatology communications, 2019). These findings should be discussed together with statements in Page 9, line 351-353, “increased fatty acid levels- as occurs in rats fed a high fat diet- activates APK in the liver and stimulates AMPK-mediated phosphorylation of ChREBP”.

2. As a general comment, this review would benefit from mentioning more translational aspects e.g. metformin in AMPK regulation of alpha-arrestin in diabetes 

Author Response

Response to Reviewer 2:
1. Reviewer 2 suggests that we consider adding the Ruderman review on the role of AMPK in the context of metabolic syndrome.

This reference has now been added (line 361)
2. Reviewer 2 felt this paper “would benefit from mentioning more translational aspects e.g. metformin in AMPK regulation of alpha-arrestin in diabetes”.

We have been unable to find any studies that show an effect of metformin on AMPK-mediated regulation of a-arrestins. It will be interesting in the future to see if any of the therapeutic benefits of metformin are derived from AMPK regulation of the arrestins.

Reviewer 3 Report

This is a well-written review of the role of alpha arrestins in the AMPK-mediated regulation of glucose transport in yeast and mammalian organisms.  The authors have written a critical review that identifies molecular mechanisms and suggests future studies that will advance our knowledge. Acceptance for publication is recommended after the authors have given consideration to the following points:

1.     In the last line of the abstract delete “~150”. 150 billion years would take us far back beyond the apparent origin of the known universe.

2.     At the end of the fifth line of the legend for Figure 1, add the word  “as” after “altered”.

3.     On line 196, change “Hxt6 [19]. Although” to “Hxt6 [19], although”.

4.     On line 204, change “is” to “its”.

5.     On line 332, change “glucose-6-phosophate” to “glucose-6-phosphate”.

6.     On line 438, change “phosho” to “phospho”.

Author Response

Reviewer 3:
1. In the last line of the abstract delete “~150”. 150 billion years would take us far back beyond the
apparent origin of the known universe.

Fixed. Billion changed to million.
2. At the end of the fifth line of the legend for Figure 1, add the word “as” after “altered”.
Done
3. On line 196, change “Hxt6 [19]. Although” to “Hxt6 [19], although”.
Done
4. On line 204, change “is” to “its”.
Done
5. On line 332, change “glucose-6-phosophate” to “glucose-6-phosphate”.
Done
6. On line 438, change “phosho” to “phospho”.
Done